# Exploring exploration with foundation agents in interactive environments

**Daniel P. Sawyer**
*Google DeepMind*
dannysawyer@google.com

**Nan Rosemary Ke**
*Google DeepMind*
nke@google.com

**Hubert Soyer**
*Google DeepMind*
soyer@google.com

**Martin Engelcke**
*Google DeepMind*
engelcke@google.com

**John Reid**
*Google DeepMind*
reidjo@google.com

**David P Reichert**
*Google DeepMind*
reichert@google.com

**Drew A. Hudson**
*Google DeepMind*
dorarad@google.com

**Alexander Lerchner**
*Google DeepMind*
lerchner@google.com

**Danilo Jimenez Rezende**
*Google DeepMind*
danilor@google.com

**Timothy P Lillicrap**
*Google DeepMind*
countzero@google.com

**Michael Mozer**
*Google DeepMind*
mcmozer@google.com

**Jane X Wang**
*Google DeepMind*
wangjane@google.com

**Reviewed on OpenReview:** `https://openreview.net/forum?id=wOrkUTrOW5`

## Abstract

Foundation models excel at single-turn reasoning, but many real-world challenges, from scientific research to technology development, require multi-turn exploration in dynamic interactive environments. Crucial components of learning from experience in these settings, such as efficiently gathering information to test hypotheses, meta-learning a model of the world's dynamics, and adapting to unexpected changes, remain largely unexplored for these models. We first evaluate foundation models in Feature World, a setting that primarily tests information gathering about a static hidden reward function. In this initial setting, we

show that state-of-the-art foundation models come close to optimal efficiency in selecting maximally informative actions in tasks with simple reward functions. As a proof of concept, we also show a model can gather information efficiently in a 3D embodied version of this task, though errors in vision limit some aspects of performance. In order to test exploration across multiple dependent turns and trials, we implement a custom, text-based version of the Alchemy environment, a benchmark designed for meta-learning. Here, agents must deduce a latent causal structure by integrating information across multiple state-dependent trials. In this more complex setting, we find that recent foundation models struggle to meta-learn strategies that enable improved performance over time. However, prompting the models to summarize their observations at regular intervals enables an emergent meta-learning process, allowing them to improve across trials. Notably, in some models, summarization also enabled adaptive re-learning of this information when the environment's rules change unexpectedly. While most models performed reasonably well on simple Feature World tasks, evaluations in Alchemy reveal stark differences in robustness among the models, with Gemini 2.5 performing best, followed by Claude 3.7, and ChatGPT-4o and o4-mini struggling the most. These results underscore Alchemy's value as a benchmark for meta-learning and strategy adaptation in foundation models. By moving beyond simple discovery to complex, stateful environments, we demonstrate that the most significant challenge for foundation agents is not selecting informative actions in the moment, but rather seeking and integrating knowledge through adaptive strategies over time. Intriguingly, we find there is likely no intrinsic barrier to future generations of foundation agents more fully mastering these abilities.

# 1 Introduction

Foundation models have demonstrated remarkable abilities in understanding and generating complex human-like text and multi-modal content (Achiam et al., 2023; Gemini Team et al., 2023; Jiang et al., 2024; Reid et al., 2024; Dubey et al., 2024; Dai et al., 2024; Deitke et al., 2024). However, this success has largely been measured in static, single-turn settings where information is provided upfront. The next frontier for these models lies in their application as interactive agents, which must operate in dynamic environments where crucial information is not given, but must be actively discovered. To achieve goals in such settings, an agent cannot merely react; it must proactively explore. This contrasts with classic reinforcement learning (RL) paradigms that use undirected exploration (Burda et al., 2018; Ecoffet et al., 2019; Badia et al., 2020). Real-world endeavors often demand a more sophisticated, hypothesis-driven approach. This involves strategically formulating beliefs about the world, designing experiments to test those beliefs, and integrating findings gathered across multiple, often state-dependent, trials. Such capabilities will become increasingly important as training on human-generated data reaches a limit and we enter the "era of experience", in which models generate their own training data through interaction with their environment (Silver & Sutton, 2025). Whether, and to what extent, today's foundation models possess this latent capacity for active exploration remains a critical and largely open question.

We evaluate LLMs in three environments: text-based and multimodal variants of Feature World, and a text-based version of Alchemy (Wang et al., 2021). The Feature World is largely stateless and does not necessitate extensive sequential decision-making, allowing us to isolate and analyze efficiency of information gathering. Alchemy, in contrast, demands strategic exploration and reasoning over multiple trials, which allows us to evaluate the foundation models' meta-learning and strategy adaptation abilities.

In this paper, we operationally define and measure three key capabilities involved in exploration: efficient information gathering, meta-learning, and strategy adaptation.

- **Efficient information gathering:** Selecting actions that maximally increase expected information gain. In Feature World, we operationally define this as the success rate ($R_{\text{success}}$) in finding a rewarding object within a fixed step budget $B$:

$$R_{\text{success}} = \frac{1}{N} \sum_{i=1}^{N} \mathbb{1}(\text{steps}_i \leq B)$$

where $N$ is the total number of episodes, $\text{steps}_i$ is the number of steps taken to find a reward in episode $i$, and $B$ is set to the maximum number of steps an optimal policy would need.

- **Meta-learning (learning to learn)**: Improving expected performance on new tasks in a given family through experience of other tasks in that family (Thrun & Pratt, 1998). In Alchemy, we measure this as the mean within-episode score improvement ($I_{\text{score}}$) between the first trial and the average of the final trials:

$$I_{\text{score}} = \frac{1}{N} \sum_{i=1}^{N} \left( \frac{1}{|T_{\text{late}}|} \sum_{t \in T_{\text{late}}} S_{i,t} - S_{i,1} \right)$$

where $S_{i,t}$ is the normalized score in episode $i$ at trial $t$, and $T_{\text{late}}$ represents the set of final trials (trials 6-10 in our experiments).

- **Strategy adaptation**: Detecting when a strategy becomes invalid due to environmental changes and adapting by learning a new one. In Alchemy, we measure this as the mean performance recovered ($S_{\text{post}}$) following an uncued change to environment dynamics:

$$S_{\text{post}} = \frac{1}{N} \sum_{i=1}^{N} \left( \frac{1}{|T_{\text{post}}|} \sum_{t \in T_{\text{post}}} S_{i,t} \right)$$

where $T_{\text{post}}$ is the set of trials following the uncued change (trials 11-20 in our experiments).

For each of the above capabilities, we say a model has that capability if the difference between the measured $R_{\text{success}}$, $I_{\text{score}}$, or $S_{\text{post}}$ for the model and the same measured quantity for a random or heuristic baseline is statistically significant ($p < 0.05$).

More specifically, this paper investigates the capacity of foundation models to conduct exploratory behavior within interactive environments in the zero-shot setting, using in-context prompting alone and without requiring task-specific training or fine-tuning.

We performed experiments using Gemini 1.5 Pro and Flash (Reid et al., 2024), Gemini 2.5 Pro and Flash (Google, 2025), Claude 3.7 Sonnet (Anthropic, 2025), and ChatGPT-4o (OpenAI, 2024) and o4-mini (OpenAI, 2025).

Overall, this work makes the following key contributions and findings:

- We conduct extensive experiments evaluating multi-turn exploration performance of foundation models across a diverse set of interactive environments. We analyzed several foundation models and a range of in-context prompting strategies, including variations in the amount of prior information and the structure of demonstrations.

- Our findings reveal a strong inherent exploratory capacity in foundation models across simple interactive settings. Specifically, all LLMs we evaluated demonstrated near-optimal performance in Feature World with simple reward functions. Likewise, some models outperformed the memoryless heuristic in Alchemy, something that the RL agents benchmarked in the original Alchemy study were unable to do.

- We find that in complex, multi-trial environments, such as Alchemy, foundation models struggle to show meta-learning (improving over trials) and strategy adaptation (re-learning a world model when the effects of actions unexpectedly change). However, both these abilities can emerge when models are prompted to summarize information across trials.

- We find stark differences in the robustness of meta-learning and strategy adaptation in frontier LLMs using Alchemy, demonstrating the utility of this environment as a benchmark for LLM exploration capabilities.

## 2 Related Work

**Exploration in RL** Information gathering is related to exploration in RL, which has been studied for tasks with sparse rewards such as Montezuma's Revenge and Pitfall in Atari, Deep-sea exploration and other tasks in the Behavior Suite for RL, and the DM-HARD-8 tasks (e.g., Burda et al., 2018; Ecoffet et al., 2019; Osband et al., 2019; Guo et al., 2022; Saade et al., 2023) as well as in unsupervised settings (e.g., Pathak et al., 2017; Guo et al., 2022). These methods commonly derive an "intrinsic" reward from the error of a predictive model (e.g., Pathak et al., 2017; Burda et al., 2018; Guo et al., 2022) or by estimating the density of visited states (e.g., Saade et al., 2023). Badia et al. (2020) use a combination of both of these types of intrinsic rewards and Tam et al. (2022) additionally use pre-trained representations. In contrast, this work relies on the prior knowledge of foundation models from internet-scale pre-training for exploration (e.g., Wang et al., 2023a; Feng et al., 2023; Lu et al., 2024b) rather than using random exploratory actions and intrinsic rewards. Also, existing RL environments (e.g., Todorov et al., 2012; Brockman et al., 2016; Tassa et al., 2018) often conflate exploration with other aspects of agent performance, making it difficult to isolate and assess a model's inherent exploratory capabilities. Such aspects include sparse or deceptive rewards and noisy, non-stationary, or multi-agent environments. We therefore chose and designed a suite of environments that allows us to systematically disentangle and control the factors influencing exploration.

**Foundation models for games** Foundation models have also been used to build agents that play games (e.g., Wang et al., 2023a;b; Feng et al., 2023; Tan et al., 2024), which often involves some form of exploration. Wang et al. (2023a) show that GPT-4 can reach impressive performance in Minecraft by incrementally building a skill library via an "automatic curriculum" stage where GPT-4 is prompted to propose novel tasks. Feng et al. (2023) prompt an LLM to explore an environment and subsequently use the collected experiences for fine-tuning the model. Unlike Wang et al. (2023a) and Feng et al. (2023), Wang et al. (2023b) and Tan et al. (2024) use image observations rather than relying on access to environment internal states. All of these works, however, focus more on improving agent performance rather than performing an explicit, systematic investigation of information gathering, meta-learning, and strategy adaptation with foundation models in controlled, zero-shot settings and in comparison to known optimal policies. LMAct benchmarks LLMs on simple games in the very long context regime and VideoGameBench tests VLMs on a collection of video games (Zhang et al., 2025; Ruoss et al., 2024). However, neither of these works investigates exploration as a capability distinct from overall performance. The BALROG benchmark incorporates a range of existing games used as RL environments and investigates exploration among a number of key capabilities (Paglieri et al., 2024). However, this is limited to a qualitative assessment and does not involve quantitative measurement of multiple clearly-defined facets of exploration as this work does.

**Exploration with foundation models** Several other works investigate exploration with foundation models, e.g., for text-based environments (Lu et al., 2024b; Huang et al., 2024), reinforcement learning from human feedback (RLHF) (Dwaracherla et al., 2024), and multi-armed bandit problems (Coda-Forno et al., 2023; Krishnamurthy et al., 2024). Unlike Krishnamurthy et al. (2024) and Dwaracherla et al. (2024) and similar to Lu et al. (2024b), this work considers stateful environments. While Lu et al. (2024b) replace components of the exploration method introduced in Ecoffet et al. (2019) with an LLM, this work studies the ability of foundation models to gather information and test hypotheses in-context via zero-shot prompting rather than using LLMs in a more modular fashion. Also adopting more modular approaches, Hu et al. (2024) use foundation models as components in a larger exploration framework and Huang et al. (2024) propose to use a smaller agent to explore the environment and a larger agent to leverage the gathered information.

**Active learning** The field of active learning (Settles, 2009) has studied how to best acquire data to improve model predictions with methods that commonly focus on highly structured data (either i.i.d. or on a graph). In contrast, this work explores efficient knowledge acquisition in more general interactive environments.

**Active embodied question answering** This work studies a similar setup to embodied question answering (EQA) (e.g., Das et al., 2018; Zhu et al., 2023; Majumdar et al., 2024; Ren et al., 2024). Similar to our work, agents in the EQA setting need to actively explore an environment to gather information. Unlike our tasks, EQA typically does not involve performing iterative experiments to infer unknown mechanisms in dynamic environments, and the optimal exploratory action typically does not depend on past observations.

**Strategy adaptation** The dorsolateral prefrontal cortex (dlPFC) is the brain region that has undergone the most reorganization in humans relative to other primates (Donahue et al., 2018). It is involved in detecting when a previously-successful strategy no longer yields expected rewards and adjusting behavioral strategy accordingly (Mansouri et al., 2007). Analogs of this capability have been studied in LLMs in the context of mental sets (Haq et al., 2025), and learning from mistakes (Tong et al., 2024). However, these studies focus on single-turn static math and reasoning problems. To our knowledge, this is the first study to investigate strategy adaptation with LLMs in the context of either exploration or interactive environments.

**AI for science** Hypothesis generation and testing is central to the scientific method and recent works research the application of foundation models in this broader domain (e.g., Romera-Paredes et al., 2023; Trinh et al., 2024; Lu et al., 2024a). Such works typically use foundation models in a highly structured protocol designed for the fixed domain in question. In contrast, DISCOVERYWORLD (Jansen et al., 2024) provides a platform with diverse, realistic tasks that test an LLM agent's ability to complete an entire scientific workflow in a range of disciplines, and ScienceAgentBench (Chen et al., 2024) evaluates agents on data-driven discovery workflows extracted from peer-reviewed literature. While such benchmarks are crucial for evaluating overall task performance and progress towards automated science, our work complements this line of inquiry by shifting the focus from *what* tasks agents can ultimately solve to *how* their underlying exploratory capabilities function and fail.

In summary, rather than assessing performance on a multi-faceted workflow, we use more abstract and controlled environments to systematically measure domain-general exploration capabilities. The key insights of our paper are therefore not about task completion in realistic domains, but about the mechanistic properties of foundation models themselves. To that end, we investigate efficient information gathering, meta-learning, and strategy adaptation across several frontier models. Among other insights, we reveal that these exploratory skills are often brittle, that the ability to learn a strategy does not imply an ability to adapt it, and that cognitive scaffolding via prompting can unlock latent meta-learning—findings that provide a more granular understanding of the fundamental challenges facing foundation agents.

## 3   Feature World

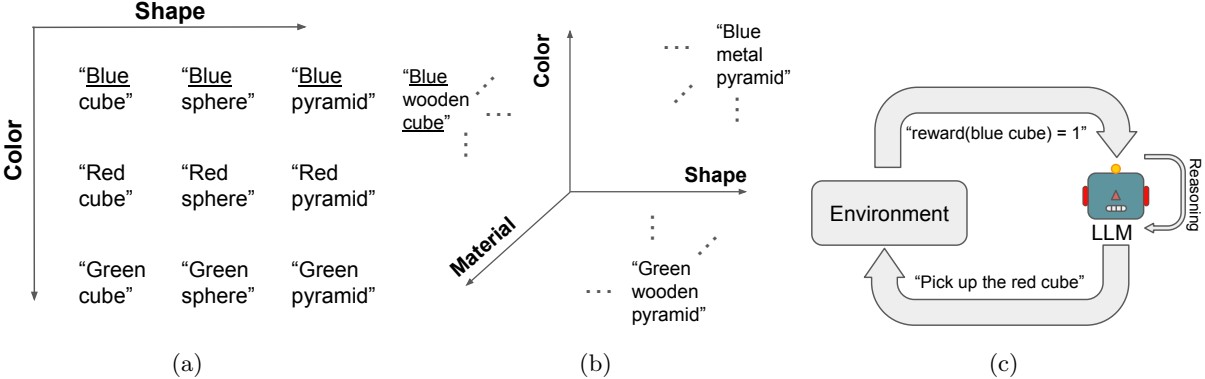

(a)                                   (b)                                   (c)

Figure 1: Task structures and experimental setups for Feature World. (a) Example task setup for text environment with single-feature reward function, with "blue" as the rewarding feature. (b) Example task setup for text environment with conjunction reward function, with "blue" and "cube" as the rewarding conjunction. (c) Schematic of text Feature World experiment setup.

We first evaluate models in Feature World: a simple, memory-less, text-based setting. In this environment, actions can be executed repeatedly without altering the underlying state dynamics due to previous actions, and each exploratory action provides immediate feedback.

### 3.1 The Feature World environments

### 3.1.1 Text environment

To investigate information gathering, we use a task where models are presented with objects possessing multiple features (e.g., color, shape). A specific feature or conjunction of features determines a reward, mirroring sparse-reward RL settings.

This task isolates multi-step information gathering within a single trial. Actions provide immediate feedback on a static reward function, with no latent dynamics to discover across trials.

To assess the capabilities of foundation models in this environment, we modulate task difficulty by adjusting two key aspects: the number of distinct colors (increasing the cognitive load) and the complexity of the reward function. Reward functions can be based on a single property (**single-feature tasks**) or a conjunction of two properties (**conjunction tasks**). See Figure 1 for a visualization of the tasks.

### 3.1.2 Proof of concept: 3D Feature World – Construction Lab

To test active exploration in a more realistic, multi-modal setting, we created a 3D version of Feature World in the Construction Lab simulation [**reference-anonymized**]. In this environment, the agent receives video input and outputs action instructions for a human to execute. This setup assesses exploratory behavior while introducing real-world challenges like visual understanding. To manage visual complexity, these 3D tasks use only three colors and a single rewarding feature, mirroring the simplest text-based condition.

### 3.2 Feature World experiments and results

Our experiments in Feature World aim to address the following two questions: (a) How does the complexity of the environment affect the information gathering efficiency of foundation models? (b) What new challenges emerge in an embodied 3D version of the task?

We evaluate on foundation models of different sizes and generations, using publicly available APIs and settings. We use the default settings for the public APIs in all cases unless noted otherwise. For Feature World, we compared ChatGPT-4o (Achiam et al., 2023; OpenAI, 2024) (200k context), Claude 3.7 Sonnet (Anthropic, 2025) (200k context), Gemini 1.5 Flash and Pro (Reid et al., 2024), and Gemini 2.5 Flash and Pro (Google, 2025) (1M context). For all experiments, we found 200k context to be sufficient.

### 3.2.1 Task setup

**Baselines** We compare to two baselines: *Optimal Baseline*: This baseline represents an upper bound on exploration performance. It selects actions that maximize information gain at each step. See Section A.2.1 for a more detailed description of the optimal strategy. *Random Baseline*: This baseline establishes a lower bound by choosing objects randomly with replacement. Both baselines are evaluated with 1000 episodes.

See Tables 1, 2, 3, and 4 in the Appendix for the prompts used in the text version of Feature World for all models.

**Evaluation** To evaluate information gathering efficiency, we assess how often models are successful at finding a rewarding object given a fixed budget of exploration steps. We set the step budget as the maximum number of steps that an optimal policy would need before finding at least one rewarding object. This measures the model's active exploration capabilities independent of its ability to draw conclusions from its observations.

For the 3D exploration task, we measure two key metrics: 1) the number of steps required to gather sufficient information to identify the reward function, and 2) the model's accuracy in correctly identifying that function from its observations. This second metric assesses the model's combined ability to efficiently explore as an actor and to draw conclusions from visual evidence.

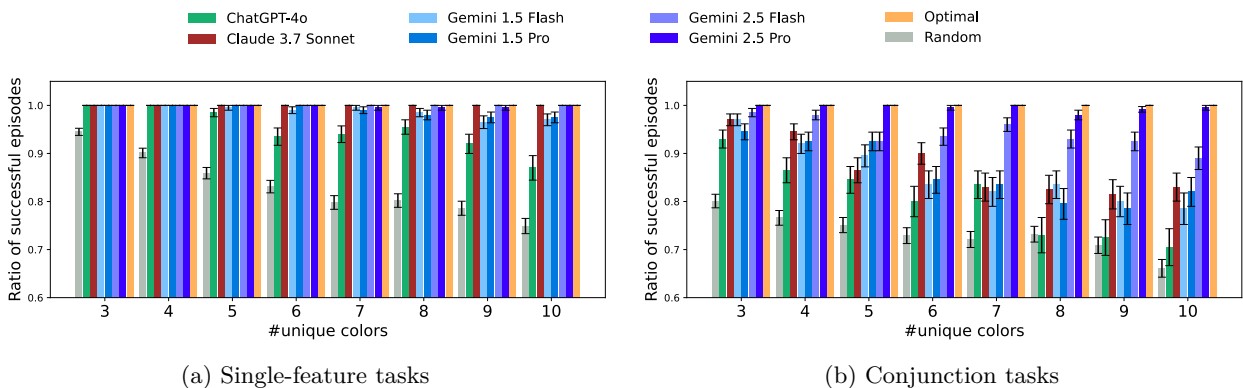

Figure 2: Fraction of Feature World episodes in which models found a rewarding object before reaching the maximum number of exploration steps. (a) Single-feature reward function. (b) Conjunction reward function. Error bars represent standard error of the mean, with 200 episodes per condition for the models and 1000 for the random and optimal baselines.

### 3.2.2 Effects of environment complexity on exploration

We examine the effect of two forms of environmental complexity on information gathering efficiency: reward function complexity and object quantity. To measure the former, we designed tasks where reward is determined by either a single feature (like "red" or "square") or a conjunction of two features (e.g., "red" and "square"). The latter requires the agent to reason about multiple properties to identify the reward-relevant combination. To investigate the impact of cognitive load on model performance, we also vary the number of unique colors in the environment.

In both single-feature and conjunction tasks, almost all models significantly outperform the random baseline in all conditions, with the exception of ChatGPT-4o, which is no better than random when the number of colors exceeds 7 in the conjunction tasks (Figure 2). This shows that most LLMs have a robust capacity for efficient information gathering.

In the single-feature task, Claude 3.7 and both sizes of Gemini 2.5 perform identically or nearly identically to the optimal baseline, with no obvious trend of decreasing performance with increasing number of colors (Figure 2a). Both sizes of Gemini 1.5 show a trend of decreasing performance with larger number of colors, but always remain close to optimal. ChatGPT-4o performance declines more quickly with number of colors, but always remains above the random baseline. This shows that, when the reward function is simple, most models are capable of nearly optimal information gathering efficiency even as cognitive load increases.

In the conjunction task, most models show a large drop in performance compared to the single feature task (Figure 2b). All models except Gemini 2.5 Pro show a substantial decrease in performance as number of colors increase, while Gemini 2.5 Pro maintained strong performance despite the increase in task difficulty. This demonstrates that reward function complexity has a large impact on information gathering efficiency for most LLMs and exacerbates the impact of cognitive load on performance.

Taken together, these results show that LLMs have a robust capacity for gathering information efficiently and generally achieve nearly optimal performance across cognitive loads for simple reward functions. On the other hand, most LLMs struggle to achieve optimal performance and degrade with increasing cognitive load when the reward function is more complex. Interestingly, our results show that Gemini 2.5 Pro is an outlier in this trend, achieving nearly optimal performance across all levels of reward function complexity and cognitive load. This shows that robust, near-optimal information gathering efficiency is possible in LLMs, and may be expected to become a common capability as other frontier models improve.

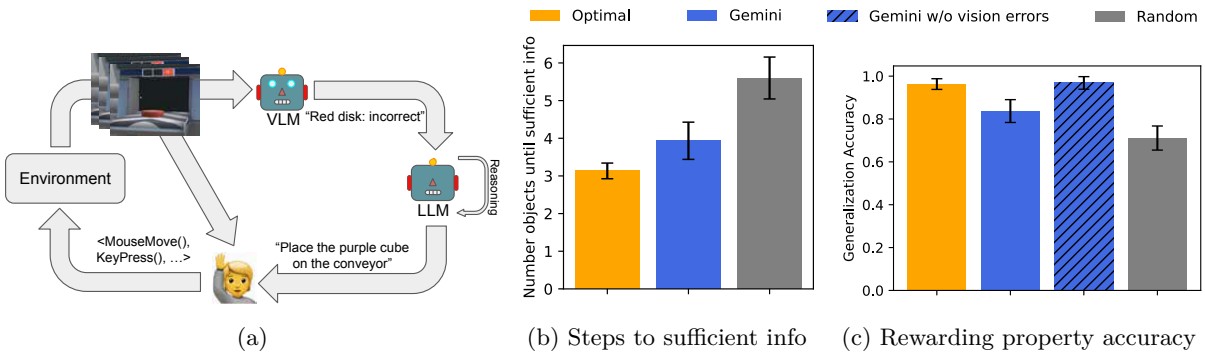

Figure 3: Schematic and performance metrics for 3D exploration task, with 15 episodes per condition. (a) Mean number of exploration steps (objects placed on the conveyor) before sufficient information is available to determine the correct factor. (b) Accuracy of the model in determining the correct rewarding feature. Hatched blue bar represents accuracy if episodes with vision errors are removed. Error bars represent standard error of the mean.

### 3.2.3 3D embodied Feature World results

As a proof of concept, we tested the multimodal and visual understanding capabilities of a model in an embodied 3D setting. See Section A.3 for more details on the setup.

We show that Gemini 1.5 Pro is capable of extracting the necessary symbolic information from video, providing action instructions in real time to a human player, and reasoning. The model's post-hoc reasoning accuracy in identifying the rewarding property was better for trajectories collected with the Gemini actor than with the random actor when trials including vision errors were removed. However, the improvement compared to the random actor in the full results that included vision errors was not statistically significant ($p = 0.13$, 2-sample t-test) (Figure 3b). 8 out of 15 of the episodes with the Gemini actor contained vision errors. Overall, the steps to sufficient information results are a proof of concept that a foundation model is capable of efficient information gathering in a 3D embodied setting requiring vision. The weaker performance due to vision errors on rewarding property accuracy suggests that multi-modal capabilities, and not reasoning, are a likely bottleneck for extending exploration with foundation models to settings closer to real-world tasks.

See Section A.3.4 for a more detailed discussion of the 3D Feature World results.

## 4 Alchemy

Following the Feature World experiments, we transition to a more complex environment that maintains a persistent but hidden task structure across multiple trials, such that an agent must take strategic action to explore in early trials to gain useful information that informs exploitative actions in later trials. For this more demanding evaluation, we selected the Alchemy environment (Wang et al., 2021). Alchemy is notable for its structured task distribution and its design to test reasoning, planning, and, importantly, exploration and meta-learning. In Alchemy, the agent needs to take actions to not only discover rewards but also latent causal dynamics, which are randomly resampled every episode. Additionally, constraints are introduced such that not all actions can be taken repeatedly within a trial (which are themselves multi-step), necessitating planning both within and across trials. These characteristics contribute to a more complex testbed for evaluating components of exploration strategies, such as meta-learning and strategy adaptation, that are measured across multiple timescales (i.e., within trials, across trials, and across episodes).

### 4.1 The Alchemy environment

Alchemy (Wang et al., 2021) is a procedurally generated environment specifically created to test meta-learning capabilities. The core gameplay involves using a set of potions to transform various visually distinctive

stones into more valuable forms, and then depositing them into a central cauldron to score points. Stone appearance varies along three feature dimensions: size, color, and shape, and their value is visually indicated by a marker. Potion effects are determined by color. A central concept in Alchemy is the "chemistry", which represents a latent causal structure that governs the value of stone appearances and the transformative effects of potions on stones. This chemistry is procedurally resampled for each episode, meaning the specific rules linking appearance, value, and potion effects change every time a new episode begins. We define a step as a single action (e.g., applying a potion), a trial as a sequence of steps ending with scoring or resource exhaustion under a fixed chemistry and finite set of objects, and an episode as a sequence of $N$ trials (defaulting to $N = 10$) where the chemistry is constant, resetting only between episodes (Figure 4a-c). Within a single episode, the agent's implicit challenge is to diagnose the current chemistry through repeated observation and experimentation. This involves operating at two timescales: making effective choices *within* each trial and synthesizing the information gathered *across trials* to learn about the latent dynamics, applying this knowledge to maximize scores in subsequent trials. After each episode, the chemistry is reset and all information from the previous episode is cleared from the model's context (with the exception of the strategy adaption experiments: see Section 4.2.3).

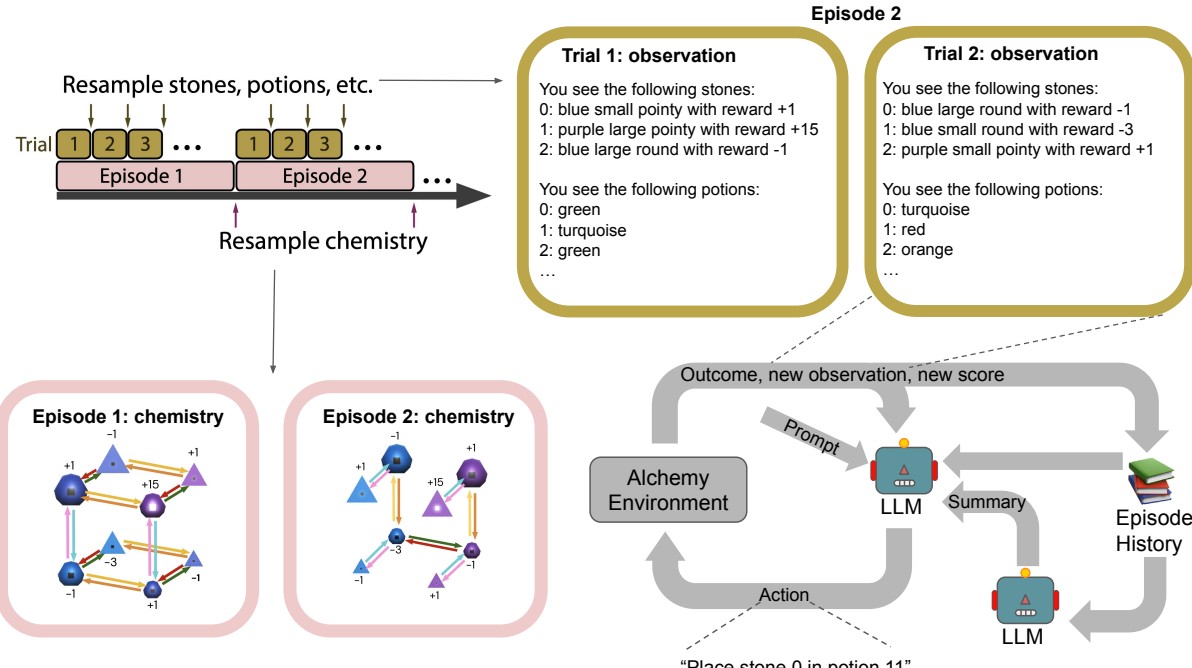

Figure 4: Task structures and experimental setup for Alchemy. **Upper left:** The structure of an Alchemy experiment. **Upper right:** Example text observations of the initial state of two separate trials from the same episode, in which stones and potions are resampled but the effects of potions and the reward values of stones remain the same. **Lower left:** Example chemistries, represented as graphs determining the effects of potions (edges) on stones of different properties (nodes), that change between episodes. **Lower right:** Directed exploration setup in which an LLM receives feedback from the environment, information from a prompt, past history of the episode, and, optionally, a summary of the episode history. The two left panels are adapted, with permission, from figures in Wang et al. (2021).

## 4.2 Alchemy experiments and results

Our experiments in Alchemy aim to address the following questions: (a) How does the multi-trial setting, which requires long-context and memory, impact the exploration performance and meta-learning of foundation models? (b) How do different prompting strategies and cross-trial summarization methods impact exploration

performance and meta-learning? (c) How well are foundation models able to adapt their learned strategies in response to uncued changes in environment dynamics?

For Alchemy, we compared Gemini 2.5 Pro (1M context), Claude 3.7 Sonnet (200k context), and o4-mini (OpenAI, 2025) (200k context), all of which are reported to employ an explicit thinking step, and ChatGPT-4o (200k context), which does not employ a thinking step. For all tasks, we found that 200k context was sufficient.

### 4.2.1 Task setup

We evaluated the LLMs on a symbolic version of Alchemy in which both actions and observations of the game state are represented as text. We used the same basic trial and episode parameters as in Wang et al. (2021).

**Baselines** We compare LLMs to two baselines: *Optimal Baseline*: the oracle baseline in Wang et al. (2021), which is a baseline that knows the underlying causal structure of the environment and can perform optimal actions. All results shown are normalized to the score of the oracle baseline. *Heuristic Baseline*: To set a baseline for reasonable performance, we use the memoryless heuristic described in Wang et al. (2021) which places random stones in random potions until either a stone reaches the maximum reward (in which case that stone is placed in the cauldron, and random selection then continues) or all stones are used up (in which case all positive-valued stones are placed in the cauldron, and the trial then ends).

**Evaluation** We assess two variables impacting the ability of models to solve the Alchemy task: 1) inclusion of prior information on invariant principles of Alchemy in the prompt (see Section A.4.1 for details), and 2) use of summarization to augment model learning across trials (see Section A.4.2 for details).

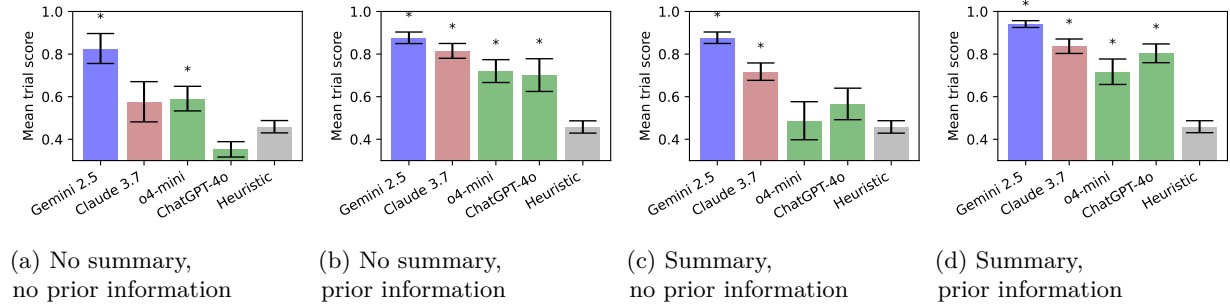

(a) No summary,
no prior information

(b) No summary,
prior information

(c) Summary,
no prior information

(d) Summary,
prior information

Figure 5: Mean Alchemy episode scores for different models and conditions. (a) No summarization, no prior information. (b) No summarization, prior information. (c) Summarization, no prior information. (d) Summarization, prior information. N=10 replicates of 10-trial episodes. Error bars represent standard error of the mean. Asterisk indicates the mean is significantly different from that of the memoryless heuristic ($p < 0.05$, paired-sample $t$-test).

We measure model performance primarily through three metrics: 1) performance: mean score over the 10 trials of an episode, 2) improvement: difference of the mean score of the last 5 trials and the score of the first trial, and 3) adaptation: the mean score for 10 trials following an unexpected change in chemistry. For all metrics, trial score is normalized as a fraction of the score of an oracle that takes the optimal set of actions for the given items. We use an additional two metrics to gain further insight into model decision making: 1) change in the number of potions used between the first trial and the last five trials (Figure 6a), and 2) the fraction of trials in which the model places at least one negative-valued stone in the cauldron. See A.4.8 for results on these latter two metrics. For all metrics, we run 10 replicate episodes with randomized chemistries.

As in Feature World, we primarily evaluate models in the zero-shot setting. We performed a small set of experiments to probe the effects of few-shot prompting, but did not find a consistent impact on performance. See Appendix A.4.5 for details.

### 4.2.2 Effects of summarization and prior information on model performance and learning

As shown in Figure 5a, Gemini and, to a lesser extent, o4-mini, significantly outperform the memoryless heuristic. Notably, the RL agents evaluated in Wang et al. (2021) did not significantly outperform the memoryless heuristic, despite being trained for 1e9 episodes. This condition with no summarization strategy and no prior information most closely mirrors the setting of the original Alchemy task experienced by the RL agents. This shows that some LLMs can act as powerful agents on tasks requiring exploitation of strategies learned through extended exploration across multiple tasks, even in environments originally designed for RL agents. However, ChatGPT-4o and Claude are not significantly better than the memoryless heuristic in this setting, and o4-mini is far from optimal.

Importantly, in Figure 6b, we see that none of the models shows a statistically significant within-episode improvement in score for this setting, suggesting that meta-learning is not operating efficiently[1].

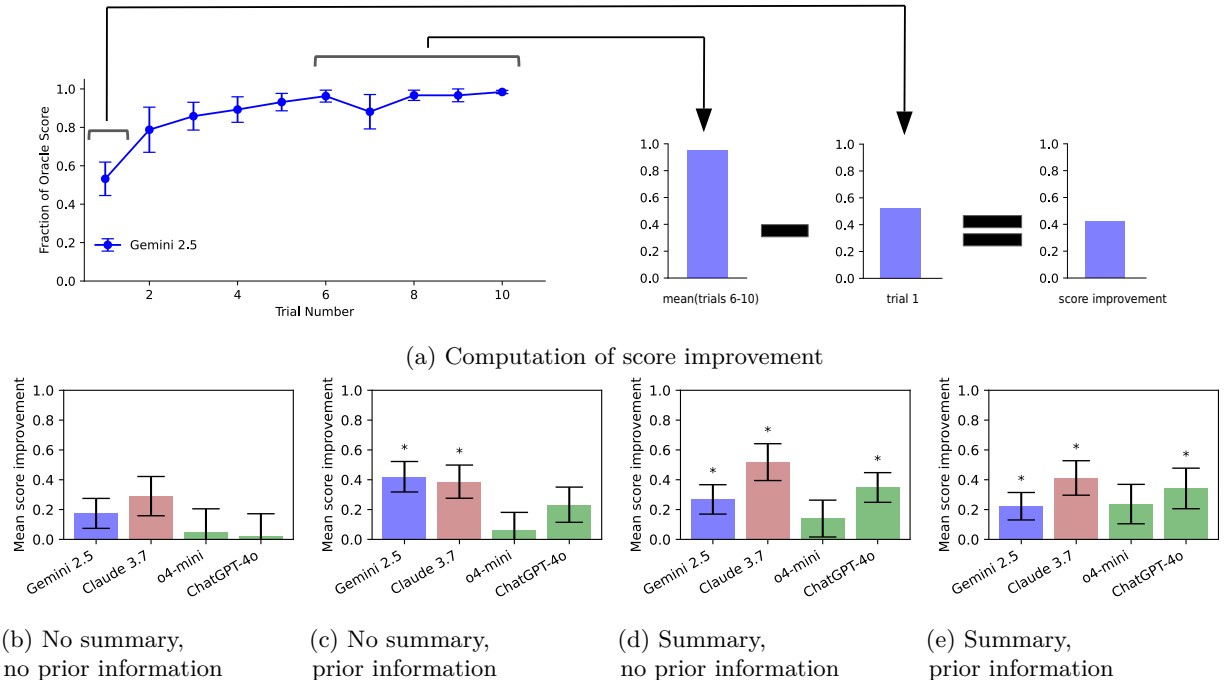

(a) Computation of score improvement

(b) No summary,
no prior information

(c) No summary,
prior information

(d) Summary,
no prior information

(e) Summary,
prior information

Figure 6: Improvement in normalized score over the episode, computed as mean of the last 5 trial scores minus the score of the first trial. (a) Illustration of how the score improvement is computed, using as an example the per-trial score trace of Gemini 2.5 in the condition with prior information but no summary. (b-e) Same conditions as Figure 5. N=10 replicates of 10-trial episodes. Error bars represent standard error of the mean. Asterisk indicates the mean is significantly different from 0 ($p < 0.05$, single-sample $t$-test).

The inclusion of prior information increased the mean scores dramatically for Claude, o4-mini, and ChatGPT-4o and slightly for Gemini (Figure 5b), and moreover resulted in score improvements over the episode becoming significant for Gemini and Claude (Figure 6). This shows that the prior information about invariant principles, despite containing no information about the specific chemistry, boosts performance and enables significant improvement over trials, supporting the hypothesis that it provides a framework for the model to generate hypotheses and targeted exploration actions[2].

---

[1]For Gemini 2.5, the lack of improvement signal appears to be due to the model learning enough information to perform well in the first trial, and then failing to improve on that. The other models both show low performance in the first trial and fail to improve.

[2]We also performed ablations on the specific types of prior information provided to determine which invariant principles were most useful to the models. See Section A.4.7 for details.

However, since Alchemy was designed specifically to evaluate models' ability to meta-learn these invariant principles from experience, the same prior information provided in the prompt ought to be present in the model's action and observation history. As such, we hypothesized that prompting the models to summarize their observations and actions after each trial would encourage them to extract equivalent information and lead to a similar boost in performance. To test this hypothesis, we implemented the summarization strategy described in Section A.4.2.

We found that summarization improved the mean scores for all models, though to a lesser extent than providing the prior information outright (Figure 5c). The boost in score improvements, however, was greater than in the prior information condition for ChatGPT-4o and Claude (Figure 6c). All models showed significant score improvement over the episode except for o4-mini. This supports the hypothesis that summarization enables meta-learning, since we would expect to see lower scores in the first trial and more improvement over time as the model acquires information over several trials rather than having it provided from the start.

To test whether the information gained in the summarization condition is functionally similar to that provided in the prior information condition, we evaluated models with both summarization and prior information. In this setting, mean scores for Claude, o4-mini, and Gemini were almost identical to the prior information condition, and the score for ChatGPT-4o increased slightly (Figure 5d). Score improvements were likewise similar to the cases with summarization only or prior information only (Figure 6d). This supports the hypothesis that summarization enabled the acquisition of information similar to or redundant with that provided in the prior information about invariant principles.

### 4.2.3 Strategy adaptation following uncued change in game rules

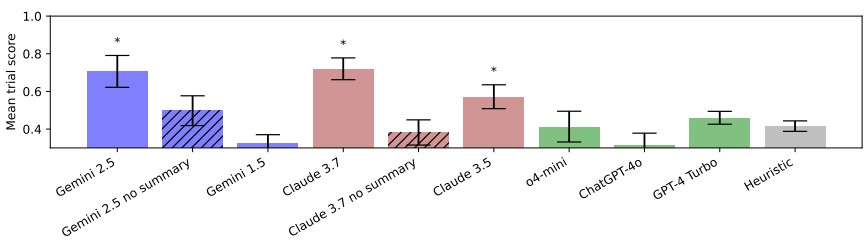

(a) Mean scores after chemistry change

Figure 7: Mean normalized model scores for trials 11-20 when an uncued change in chemistry occurs halfway through a 20-trial episode. Hatched pattern represents no summary. Error bars represent standard error of the mean, across 10 replicates.

To evaluate strategy adaptation, a human ability associated with dorsolateral prefrontal cortex (dlPFC) function (Mansouri et al., 2007; Donahue et al., 2018), in LLMs, we modified the task setup such that the models are exposed to two consecutive episodes before their observation history is cleared. However, the change in chemistry in the transition to the second episode occurs silently, leaving the model to grapple with unexpected outcomes of previously-predictable actions. We ran experiments with summaries enabled and reward and potion pair information provided, but withheld causal information since it was no longer accurate given the change in chemistry. See Figure 11 in the appendix for full timeseries plots of these results.

Of our four thinking models studied, Gemini 2.5 and Claude 3.7 were able to regain full performance after an initial drop following the change in chemistry (Figure 7, Figure 11a,b). However, o4-mini and ChatGPT-4o were indistinguishable from the heuristic policy following the change (Fig 11c,f). Unlike o4-mini, ChatGPT-4o had strong performance and improvement over trials prior to the change, showing that the ability to improve through learning of an initial strategy (meta-learning) does not necessarily predict the ability to learn a new strategy when the world changes (strategy adaptation).

To test whether the recent generation of models with thinking capability are required for successful strategy adaptation, we also tested three previous generation models without thinking capability: Gemini 1.5 Pro, Claude 3.5 Sonnet, and GPT-4 Turbo. While Gemini 1.5 and GPT-4 Turbo were indistinguishable from

random in both episodes (Figure 11g,h), Claude 3.5 showed successful strategy adaptation (Figure 11i), demonstrating that thinking or other features of recent models are not strictly necessary for this capability, though they likely help.

To test whether Gemini 2.5 and Claude 3.7 have native strategy adaptation ability without augmentations, we evaluated both models with summary and prior information disabled. In both cases, the models showed a collapse in performance following the chemistry change (Figure 11d,e). This shows that even models with strong native exploration abilities struggle with strategy adaptation when not provided with task-specific prompt augmentations.

Overall, these results show that while strategy adaptation is not a robust native capability of LLMs, it can emerge in some state-of-the-art LLMs with proper prompt augmentations. This suggests that there is no fundamental barrier to LLMs employing strategy adaptation.

### 4.3 Alchemy conclusions

Taken together, our results show that integrating information over long time horizons through adaptable strategies in context is a frontier challenge in LLMs. In particular, we show that even the strongest models are generally poor at meta-learning and strategy adaptation without task-specific augmentations. However, the emergence of these skills after prompting for summarization suggests that LLMs have a latent ability to improve and adapt through exploration. Likewise, the comparatively strong performance and adaptation abilities of Gemini 2.5 and Claude 3.7 relative to other models and previous versions suggest that deficits in exploration ability can vanish as a consequence of more general model improvements.

## 5 Discussion and limitations

This work provides critical insights into the active exploration capabilities of foundation models. In Feature World, we find that exploration efficiency remains very close to optimal for most models on tasks with single-feature reward functions, and for at least one model with more complex reward functions.

While Gemini 1.5 Pro showed efficient information gathering in the 3D Feature World, as well as the ability to draw conclusions about the environment dynamics (when vision errors were excluded), the experiments also underscored that accurate visual interpretation and multi-modal processing can present significant challenges, potentially bottlenecking performance more than reasoning capabilities alone.

We show Alchemy is a challenging benchmark for meta-learning and strategy adaptation, where foundation models struggle without prompt augmentations. Critically, we find that inter-trial summarization unlocks these abilities, suggesting they are latent capabilities that are not fundamentally out of reach for future models.

While recent benchmarks like DISCOVERYWORLD (Jansen et al., 2024) and ScienceAgentBench (Chen et al., 2024) evaluate agents on realistic, end-to-end scientific workflows, our work complements this macro-level view by isolating and quantifying the specific cognitive mechanisms underlying exploration in controlled, abstract settings. Our findings refine the broad observation in these benchmarks that agents struggle with discovery tasks by identifying precisely where this struggle occurs. In Feature World, we find that foundation models act as near-optimal information gatherers, demonstrating that the immediate logic of selecting informative actions—a core component of experimental design—is already a robust capability, contradicting the impression from broader benchmarks that agents inherently lack exploratory capability. Instead, our Alchemy results suggest that some of the failure modes observed in broader benchmarks may stem from deficits in the ability to integrate observations into a coherent and adaptable world model across trials. Furthermore, we demonstrate that this deficit is not permanent: lightweight scaffolding, such as periodic summarization, can unlock these latent capabilities.

One limitation of this study is that we exclusively evaluated zero-shot performance using in-context learning. While this isolates the inherent capabilities derived from pre-training and prompting, it does not explore how performance might change with task-specific fine-tuning or other adaptation methods. However, while finetuning is closely related to in-context or meta-learning, we believe that a direct comparison of the two

constitutes an important but distinct topic of study and thus is outside the scope of the current work. This topic has been investigated in papers comparing and contrasting finetuning with in-context learning, e.g., Chan et al. (2022), Mosbach et al. (2023), Chan et al. (2024).

Another limitation is that our 3D embodied Feature World experiments are a preliminary proof-of-concept with a limited scope, involving a single model in a low-complexity setting. Furthermore, by using a human-in-the-loop for motor control, we intentionally abstracted away the challenges of action generation and physical grounding. A full assessment of embodied foundation models would require integration with, or generation of, motor policies.

Overall, we demonstrate that the frontier of autonomous exploration lies in complex, multi-trial environments where models must continuously integrate information to meta-learn and adapt a world model. While challenging, these abilities can be elicited with prompt augmentations like summarization, suggesting no intrinsic barrier to their emergence. Benchmarks like Alchemy are crucial for testing these capabilities as models improve and we enter the "era of experience" (Silver & Sutton, 2025).

## Broader Impact Statement

This work primarily benchmarks and analyzes the performance of existing models, rather than introducing new models or methods. While our finding that summarization enables meta-learning in models could be used to improve the autonomy of LLM agents, the focus is on characterization of different models and behaviors given our specific settings and benchmarks. For these reasons, we expect the primary impacts of our work to be an improved understanding of the existing capabilities of LLM agents and we do not anticipate any negative societal impacts for the research in this paper.

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

# A  Technical Appendices and Supplementary Material

## A.1  Prompts for Feature World

Prompts used for the foundation models in Feature World. Prompts are provided verbatim, with the exception of newlines added to fit the text within the table boundaries.

| Task | Prompt |
|---|---|
| Text Environment Single Factor Task | You are playing a text-based game. Your goal is to find which object property leads to a non-zero reward in as few steps as possible.

Game Rules:
- There are objects with different colors and shapes.
- Picking up an object gives you a reward of either 0 or 1.
- There is a single property, i.e., one particular color OR shape, that leads to a reward of 1.

{scene_description}

{action_reward_description}

Respond with this format, please be specific about the object:

* Action: pick up <colored> <object>
* Stop: <YES> or <NO>
*
* Which factor influence reward? <COLOR> or <SHAPE>  or <UNSURE>
* WINNING COMBINATION: <State the specific color or shape that leads to reward>

Explain your reasoning thoroughly. |

Table 1: In-context prompt used for the text Feature World environments for the single-feature tasks.

| Task | Prompt |
|---|---|
| Text Environment Multi Factor Task | You are playing a text-based game. Your goal is to find which combination of object properties leads to a non-zero reward in as few steps as possible.

Game Rules:
- There are objects with different colors, shapes, and textures.
- Picking up an object gives you a reward of either 0 or 1.
- There is a single combination of two properties, i.e., a color and shape, a shape and texture, or a color and texture, that leads to a reward of 1.

{scene_description}

{action_reward_description}

Respond with this format, please be specific about the object:

* Action: pick up <colored> <textured> <object>
* Stop: <YES> or <NO>
*
* Which combination of factors influence reward? <COLOR, SHAPE> or <COLOR, TEXTURE> or <TEXTURE, SHAPE>  or <UNSURE>
* WINNING COMBINATION: <State the specific combination of properties (e.g., color and shape, shape and texture, or color and texture.>

Explain your reasoning thoroughly. |

Table 2: In-context prompt used for the text Feature World environments for the conjunction tasks.

| Task | Prompt |
|------|--------|
| Text Environment Single Factor Task Structured JSON | You are playing a text-based game. Your goal is to find which object property leads to a non-zero reward in as few steps as possible.

Game Rules:
- There are objects with different colors and shapes.
- Picking up an object gives you a reward of either 0 or 1.
- There is a single property, i.e., one particular color OR shape, that leads to a reward of 1.

{scene_description}

{action_reward_description}

Your response must conform to the following JSON format:
{{
    "next_object_picked_up": What object should be picked up next? This should be in format <COLOR> <SHAPE>.
    "stop": YES or NO, do you want to end the game after this step?
    "rewarding_factor": What factor do you think most is related to reward? <COLOR> or <SHAPE> or <UNSURE>
    "winning_combination": State the specific color or shape that leads to reward. If you're not sure, it's okay to say <UNSURE>
}} |

Table 3: In-context prompt used for the text Feature World environments for the single-feature tasks, using a structured JSON output format[3].

| Task | Prompt |
|------|--------|
| Text Environment Multi Factor Task Structured JSON | You are playing a text-based game. Your goal is to find which combination of object properties leads to a non-zero reward in as few steps as possible.

Game Rules:
- There are objects with different colors, shapes, and textures.
- Picking up an object gives you a reward of either 0 or 1.
- There is a single combination of two properties, i.e., a color and shape, a shape and texture, or a color and texture, that leads to a reward of 1.

{scene_description}

{action_reward_description}

Your response must conform to the following JSON format:
{{
    "next_object_picked_up": What object should be picked up next? This should be in format <COLOR> <TEXTURE> <SHAPE>.
    "stop": YES or NO, do you want to end the game after this step?
    "rewarding_factor": What factors do you think most are related to reward? <COLOR, SHAPE> or <COLOR, TEXTURE> or <TEXTURE, SHAPE>  or <UNSURE>
    "winning_combination": State the specific combination of properties (color and shape, shape and texture, or color and texture) that leads to reward. If you're not sure, it's okay to say <UNSURE>
}} |

Table 4: In-context prompt used for the text Feature World environments for the conjunction tasks, using a structured JSON output format.

| Task | Prompt |
|---|---|
| 3D Environment Iterative Exploration: vision | You are an expert video game player who is annotating videos of gameplay.

In this game, the player controls a robot in a factory room, which contains objects of various shapes and colors, such as red planks, blue cubes, green cylinders, orange disks, yellow pyramids, etc.
The player can pick up and move objects using a blue laser beam.
The player is trying to place the correct type of object on the conveyor belt.
If the object is correct, the object disappears in the machine and the light on the machine turns green.
If the object is incorrect, the light on the machine turns red and the object is pushed off.

The possible colors are red, green, blue, yellow, purple, and orange.
The possible shapes are cylinder, cube, plank/board, pyramid, and disk.

Your goal is to accurately and comprehensively list every object that the player places on the input conveyor belt, along with the timestamp of when the object was placed and whether the object is correct or incorrect.

Your response should be in the following format:
0 [timestamp 0] <1st object placed on conveyor> : <correct / incorrect>
1 [timestamp 1] <2nd object placed on conveyor> : <correct / incorrect>
2 [timestamp 2] <3rd object placed on conveyor> : <correct / incorrect>
3 [timestamp 3] <4th object placed on conveyor> : <correct / incorrect>
... |
| 3D Environment Iterative Exploration: reasoning | Now we want to explain how this game works.
The goal of the game is to place all objects with the right property, such as a particular color or shape, on the conveyor belt.
Let's try to find the next action to take to figure out what factor (color or shape) determines the correctness of the object.

If there is no history of objects yet, tell the player to pick up a random object you can see in the room from the video.
If you have no video input yet, tell the player to explore the room.
Otherwise, follow the instructions below.

  Important: You have VERY FEW turns left. Choose your next action carefully to maximize information.

  Think step-by-step:

  1. What pattern do you see in the correct objects so far?
  2. **Consider which colors and shapes have NEVER been correct. This eliminates BOTH the color AND shape from being correct.**
  3. What color or shape seems MOST promising to test next?
  4. Why will this choice give you the most useful information, even if it isn't a correct object?

  Explain your reasoning thoroughly. Don't just guess! Each turn is precious.

  After doing your reasoing, respond at the end with this format, please be specific about the object:

  * CORRECT PROPERTY: <COLOR> or <SHAPE> or <UNSURE>
  * NEXT COMMAND: place the <colored> <object> on the conveyor belt. |

Table 5: In-context prompts used for the 3D Construction Lab environment in the exploration phase for the Gemini agent.

| Task | Prompt |
|---|---|
| 3D Environment Trajectory Review: vision | You are an expert video game player who is annotating videos of gameplay.

In this game, the player controls a robot in a factory room, which contains objects of
various shapes and colors, such as red planks, blue cubes, green cylinders,
orange disks, yellow pyramids, etc.
The player can pick up and move objects using a blue laser beam. The player
is trying to place the correct type of object on the conveyor belt. If the object
is correct, the object goes through and the light on the machine turns green.
If the object is incorrect, the light on the machine turns red and the object
is pushed off.

The possible colors are red, green, blue, yellow, purple, and orange.
The possible shapes are cylinder, cube, plank/board, pyramid, and disk.

Your goal is to accurately and comprehensively list every object that the
player places on the input conveyor belt, along with the timestamp of when the object was placed
and whether the object is correct or incorrect.

Your response should be in the following format:
0 [timestamp 0] <1st object placed on conveyor> : <correct / incorrect>
1 [timestamp 1] <2nd object placed on conveyor> : <correct / incorrect>
2 [timestamp 2] <3rd object placed on conveyor> : <correct / incorrect>
3 [timestamp 3] <4th object placed on conveyor> : <correct / incorrect>
... |
| 3D Environment Trajectory Review: reasoning | Now we want to explain how this game works.
The goal of the game is to place all objects with the right property, such as a particular color or shape,
on the conveyor belt.

Based on the observations above of which objects were placed on the conveyor belt
and which ones were correct or incorrect, explain your reasoning and state what the right object
property is.
The right property is either a specific shape or a specific color.

Your response should be in the following format:
REASONING: <Explain your reasoning for how you deduced the right object property.>
TARGET PROPERTY: <State what the specific correct shape OR specific correct color is.> |
| 3D Environment Trajectory Review: generalization | Based on what you determined the correct object property to be, state whether
    each of the following objects would be correct if placed on the conveyor belt: |

Table 6: In-context prompts used for the 3D Construction Lab environment in the review phase for all agent conditions.

### A.2 Additional results - Feature World

### A.2.1 Description of optimal strategies

To illustrate the optimal strategy, consider a task where the hidden rewarding property is "red." The strategy unfolds in two phases. The first phase is exploration, where the goal is to find a successful object by maximizing information gain. If an attempt on a "blue toy" fails, the agent learns that *if* color is the rule, "blue" is not the answer, and *if* shape is the rule, "toy" is not the answer. The optimal next action is to test an object with entirely new features, like a "yellow sphere", to efficiently explore the remaining possibilities.

Once an action succeeds—for example, picking up a "red box"—the strategy shifts to the second phase: isolation. The agent must now disambiguate whether "red" or "box" is the true cause. The optimal way to do this is to test a new object that changes only one of these features, such as a "red sphere" or a "green box", to definitively pinpoint the rewarding property.

### A.3 Exploration in 3D embodied environments: Construction Lab

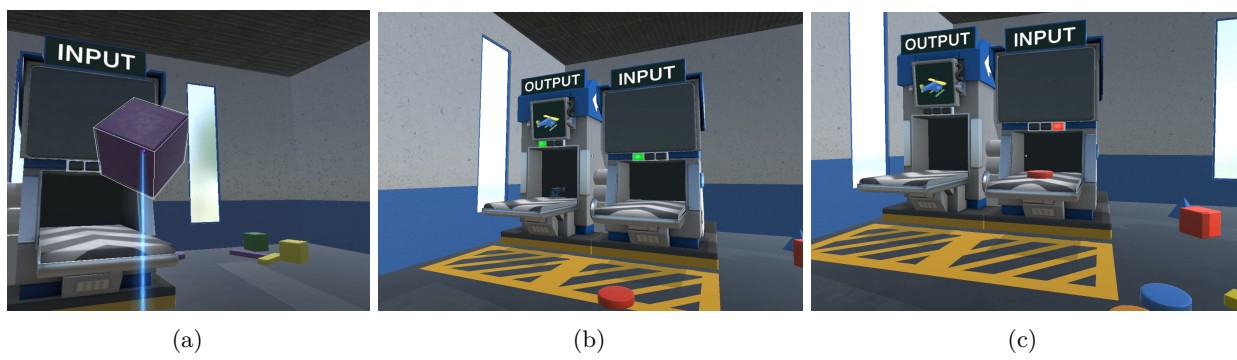

(a)    (b)    (c)

Figure 8: Screenshots of gameplay in the 3D Construction Lab environment. (a) The agent uses a blue laser beam to pick up objects. (b) Result of a correct object placement. (c) Result of an incorrect object placement.

To further evaluate the foundation models in a 3D embodied environment, we implement an analogous task to the text-based environment in a factory-style simulation called Construction Lab. Construction Lab was introduced in [**reference-anonymized**] as a simulation environment that includes both game-like mechanics and simplified but non-trivial object manipulation and physical reasoning.

In this work, we focus on a task that requires the player to operate a simple machine called the Exchanger. The Exchanger requires objects with specific properties to be placed on an input conveyor belt (Figure 8). If an object matches the requirement, the input is consumed, a green light shows for a few seconds, and an output object is produced on an output belt. If the object is invalid, the machine rejects it by reversing the input belt and a red error light is activated. No cues are provided regarding the correct input object required, and thus the task entails determining what the correct object properties are through trial and error, observing how the machine responds to input objects, and drawing appropriate inferences.

Through the use of this 3D, visually rich environment that mirrors the challenges of the text-based environment, we are able to investigate the effects of visual complexity on active information gathering, reasoning, and hypothesis testing.

### A.3.1 Task setup

A number of additional challenges must be addressed when performing this exploration task in a 3D embodied environment. First, the agent must assess both the current state of the environment and the consequences of any actions taken through vision. Second, the agent requires a motor control module to execute exploratory actions. We use Gemini 1.5 Pro's multi-modal functionality to ingest video input from Construction Lab sub-sampled to 1.5 Hz and 320 x 240 resolution. To disentangle vision and reasoning performance from

translation of natural language instructions into a complex keyboard-and-mouse action space, we adopt a setup in which instructions are provided to a human actor who performs the exploratory actions online.

We assess Gemini 1.5 Pro's ability to generate these exploratory instructions by comparing against an optimal and random baseline, mirroring those in the text environment. The optimal strategy was performed by a single human performing the task according to an optimal policy that maximally reduced uncertainly about the correct property. The random strategy was performed according to a policy that selects a random object from the room at each step, with replacement.

As running the 3D environment and using human actors in the loop reduces experimental throughput, we limit ourselves to a single level of environment and reward function complexity. We choose the condition with 3 colors and 1 causal factor, as conditions with more colors had significant visual clutter. Each task is randomly generated as follows: at the beginning of the episode, 3 unique colors and 3 unique shapes are randomly selected from 6 colors and 5 shapes, and objects with each shape-color combination are placed in random locations in the environment, for a total of 9 objects. One property, either a shape or a color, is randomly selected as the correct property, for a total of 3 correct objects. The player and the Exchanger machine with input and output conveyor are likewise placed randomly in the room. A gameplay episode ends when either all 3 correct objects are placed on the input conveyor or 2 minutes have elapsed.

### A.3.2   Gemini-based agent

The Gemini agent is implemented as follows: every 10 seconds, the model is fed the most recent 100 video frames (or 67 seconds) of gameplay and queried in two stages, during which gameplay is paused for the human actor. We implement a two-stage procedure with a vision stage and a reasoning stage, which we found improves accuracy for each stage compared with running both together. In the first stage, Gemini is asked to list, for every object placed on the input conveyor, the timestamp at which it was placed, its color and shape, and whether it was correct or not (as indicated by a red or green light on the machine). In the second stage, Gemini is provided the output of the first stage (subsequent video frames and list of objects placed with their reward values) and prompted to select a next exploratory action to maximize information gain, similar to the text environment. The human actor is provided only with the command generated by the second stage, such as "place the red cube on the conveyor."

All video trajectories are processed in the same way, regardless of how the exploration instructions were generated. Specifically, we truncate the video to include only the first 4 object attempts. Gemini is then called on the truncated video in three steps: vision, reasoning, and generalization. In the vision step, it is asked to list all the objects placed on the conveyor and whether they were correct, similar to the vision step in the exploration policy. In the reasoning step, it is asked to deduce the correct object property based on its observations. In the generalization step, it is asked to predict whether each object in a list of hypothetical objects would give a reward. See Appendix A.1 for specific prompts used.

### A.3.3   Evaluation

To evaluate different aspects of performance for each agent type, we measure relevant property accuracy and number of objects until sufficient information is acquired to determine the correct property, assuming perfect reasoning. We also record the number of vision errors made by the VLM when listing objects in the full video, defined as misclassifying the shape, color, or correctness of an object placed on the conveyor, or omitting mention of an object placed on the conveyor. Because internal game states are not exposed in our experiments, we use manual human annotation of video trajectories to collect the above metrics and error counts. We collect a total of 15 trajectories for each agent type.

### A.3.4   Results

In the exploration efficiency metric, we see the same trends in the results for the 3D embodied environment as for the text environment, with Gemini's exploration efficiency significantly outperforming the random baseline and approaching the optimal baseline (Figure 3a). These results suggest that the additional complexity of an imperfect vision system and partially observed environment state are not significant limitations in generalizing

directed exploration capabilities to embodied 3D environments. In the accuracy metric (Figure 3c), the picture is more nuanced. For relevant property accuracy, the difference between performance with the Gemini agent and the random agent was not statistically significant ($p > 0.05$, paired sample t-test).

This result is interesting because VLM vision is also necessary for the exploration phase, where there was no discrepancy in performance. A likely reason for this is that the iterative nature of the exploration task makes it robust to occasional errors. Because the model must re-list all objects placed at each step, chance errors made during one step do not propagate to later steps.

To probe the reason for the gap in accuracy performance, we also computed results where we filtered out trajectories in which the vision step made an error (Figure 3b). In these results, accuracies for the Gemini and optimal agents are nearly identical and their differences with the random agent are statistically significant ($p < 0.05$, two sample t-test). These results suggest that errors in the vision step, rather than reasoning or exploration, are responsible for the relatively reduced accuracy in the Gemini agent condition.

Taken together, results in the Construction Lab show that the directed exploration capabilities of foundation models robustly generalize from text-based environments to embodied 3D environments, though overall accuracy of the system is somewhat reduced by imperfect performance of the VLM's object and action recognition in videos. This indicates that the challenges of multi-modal reasoning from realistic simulated video could be addressed by focusing on the vision and action recognition capabilities of foundation models separately from their reasoning capabilities.

### A.4  Additional Alchemy details and results

### A.4.1  Invariant principles in Alchemy

While the specific chemistry changes per episode, Alchemy includes invariant principles or abstract regularities that span all episodes. These invariants are crucial because discovering and exploiting them over many episodes is the essence of the meta-learning problem in Alchemy.Such invariant properties include, among others: 1. within an episode, stones with the same visual features have the same value and respond identically to potions, and potions of the same color likewise have the same effects; 2. potions come in fixed pairs (e.g., red/green, yellow/orange, pink/turquoise) which always have opposite effects; 3. the underlying causal graph topology is structured by a generative grammar, though some edges might be missing, creating "bottlenecks"; and 4. the maximum stone score is 15, and the minimum score is -3. It's important to note that all of our experiments were conducted without any post-training, and so all models are presumed to lack knowledge of these invariant task structures.[4] We therefore conduct experiments to test the effects of including various components of these invariant properties into the prompt. This allows us to disentangle the challenge of acquiring such meta-learned knowledge from that of already possessing it and being able to use it to inform smart exploration.

In the case with *no prior information*, the model is given the main prompt to introduce Alchemy, which describes the general gameplay mechanics (Figure 9). We hypothesized that, in order to learn which potions optimally improve stone reward value for a given combination of stone properties, the models require an understanding of the invariant properties of the Alchemy tasks, which provide a framework on which the model can integrate evidence from its observations. To test this hypothesis, we provided additional information about the reward, potion pairing, and causal mechanics of the game (*prior information* condition, Figure 9).

### A.4.2  Cross-trial summarization

The multi-trial structure of Alchemy introduces a need to manage very long contexts and perform inference across multiple timescales and abstraction—a challenge that often arises in real-world, and particularly multi-modal, settings. Even in the case that it is possible to fit all observations, actions, and rewards from a multi-turn interaction within the model's context window, it is unclear whether current models are capable of natively managing ever-growing context while maintaining inference and reasoning capabilities. We therefore

---

[4]Because the Alchemy environment was published in 2021, knowledge of it may be included in the pretraining data for the foundation models studied here. However, when we probed the models, we found they entirely lacked knowledge about potion pairs and min/max rewards and had limited knowledge of the environment in general.

Main prompt $\Bigg\{$ You are playing a text-based game called Alchemy where you place stones of different shapes (round or pointy), sizes (small or large), and colors (blue or purple) into potions that change the stones' properties. The goal is to maximize the reward value of the stones, and then place them into the cauldron to increase the total score. Placing a stone in the cauldron adds the current reward value of that stone to the total score. However, there might be more rules to the game than mentioned here.

Reward information $\Big\{$ The maximum stone reward value is 15 and the minimum is -3.

Potion pairing information $\Big\{$ There are six different potion colors, which come in three pairs: yellow/orange, red/green, and pink/turquoise. Potions in a pair have the opposite effect from each other on a given property.

Causal information $\Big\{$ A stone's reward value is determined by its properties. The game is deterministic: a potion of a given color always has the same effect on a stone with given properties, and a stone with given properties always has the same reward value.

Figure 9: Prompt and different components comprising the *prior information* (reward information, potion pairing information, and causal information).

examined and along-context summarization method. We hypothesized that summarization would offer a benefit due to the multi-trial structure of the task that requires extended directed exploration and inference across multiple timescales.

Specifically, the summary is constructed at the end of each trial by asking the model to create a summary conditioned on the current trial's events as well as the summary from the previous trial. At each step, the model is asked to take actions based on the summarization as well as the current trial's events so far. We additionally have a *no summarization* condition, in which all episode events up until the current step are included in the context.

### A.4.3 Alchemy experimental setup details

We implemented a custom version of symbolic Alchemy, modified so that observations and actions are given in natural language format. Episode chemistries are sampled every episode and held constant for the duration of 10 trials, although new stones and potions are resampled every trial. Trials last for a maximum of 20 steps, and the agent could choose to end the episode at any step. At each step, the agent could choose to select a stone and then either a potion or the cauldron (to cache the stone and obtain the reward), or to end the trial. See Figure 4 (d) for a visualization of the tasks.

We considered the condition of procedural generation in which the transition graph has random bottlenecks: This is the hardest version of the task in which certain transformations (e.g., the effect of a blue potion) are only possible if specific initial conditions of the stone are met (e.g., the stone being transformed is large and round).

### A.4.4 Example model outputs in Alchemy

Below, we display examples of raw outputs of Gemini 2.5 from trail 1 and trial 5. For each trial, we show 1) the model's reasoning in step 1 of the trial, 2) the model's reasoning in step 5 of the trial, and 3) the model's summary of the trial. All outputs are drawn from the condition with summaries but no prior information.

For the reasoning steps, we also display the observation of the game state provided to the model by the environment, labeled GAME STATE UPDATE.

| Condition | Output |
|---|---|
| Trial 1 Step 1 | GAME STATE UPDATE:
New episode
New trial
New observation
You see the following stones:
0: purple small pointy with reward -1
1: blue small pointy with reward 1
2: purple large pointy with reward 1
You see the following potions:
0: pink
1: turquoise
2: turquoise
3: pink
4: yellow
5: red
6: yellow
7: red
8: red
9: turquoise
10: pink
11: orange

Current trial score: 0

MODEL RESPONSE:
OBSERVATION: This is the first action of the trial. I have not yet used any potions, so their effects are unknown.
REASONING: My initial goal is to understand the effects of the different potions. I will begin by testing the first available potion, which is the pink potion at index 0. I am choosing to use stone 0 (purple small pointy, reward -1) for this experiment. Since this stone has a negative reward, there is little risk in modifying it, and any positive change would be a significant gain. This action will reveal what pink potions do to purple, small, and pointy stones.
ACTION: Place stone 0 in potion 0. |
| Trial 1 Step 5 | GAME STATE UPDATE:
What stone do you use? 1
What potion do you use? 5
Outcome stone: purple small pointy with reward -1
New observation
You see the following stones:
0: purple large pointy with reward 1
1: purple small pointy with reward -1
2: purple large pointy with reward 1
You see the following potions:
0: None
1: None
2: None
3: pink
4: None
5: None
6: yellow
7: red
8: red
9: turquoise
10: pink
11: orange

Current trial score: 0

MODEL RESPONSE:
OBSERVATION: Placing stone 1 (blue small pointy, reward 1) into the red potion at index 5 changed its color from blue to purple. This also inverted its reward value from 1 to -1. The stone is now a purple small pointy stone.
REASONING: The last action revealed that red potions change blue stones to purple and reduce their reward. My new stone 1 is now a purple small pointy stone with a reward of -1, which is exactly what stone 0 was at the start of the trial. In step 2, I learned that a turquoise potion changed stone 0 from a small purple stone to a large purple stone, increasing its reward from -1 to 1. I will now attempt to replicate this effect on the new stone 1 using the turquoise potion at index 9. This will both confirm my hypothesis about turquoise potions and improve the value of a stone. |
| Trial 1 Summary | ### Potion Effects
* **Pink Potion:** Appears to have no effect. It was tested on a `purple small pointy` stone with no change.
* **Red Potion:** Changes a stone's color from `blue` to `purple`. It does not affect `purple` stones. The effect on the reward depends on the stone's size: the reward is inverted for `small` stones but stays the same for `large` stones.
* **Turquoise Potion:** Changes the size of `purple small` stones to `purple large`. It has no effect on `blue` stones or already `large` stones. The reward is inverted.
* **Yellow Potion:** Changes a stone's shape from `round` to `pointy`. It has no effect on `pointy` stones. The reward is inverted.
* **Orange Potion:** Changes a stone's shape from `pointy` to `round`. It has no effect on `round` stones. The reward is inverted.
### Highest Reward Combination
The highest reward value seen in the game is **+1**. Based on the potion effects, the combinations of properties that result in this highest reward are:
* `blue small pointy`
* `purple large pointy`
* `blue large pointy`
* `purple small round` |

Table 7: Outputs of Gemini 2.5 for Trial 1 of Alchemy in the condition with summarization and no prior information.

| Condition | Output |
|---|---|
| Trial 5 Step 1 | GAME STATE UPDATE:
New observation
You see the following stones:
0: blue small round with reward -1
1: purple large round with reward -1
2: blue large pointy with reward 15
You see the following potions:
0: orange
1: orange
2: red
3: pink
4: yellow
5: red
6: red
7: green
8: red
9: red
10: orange
11: pink

MODEL RESPONSE:
OBSERVATION: The game has started with one stone (Stone 2) already in the 'blue large pointy' state, which I have identified as the highest possible reward state (+15).
REASONING: Stone 2 currently has a reward of 15. Based on my previous findings, this is the maximum achievable reward. Using any potion on it would risk lowering its value. Therefore, the most logical first step is to place it in the cauldron immediately to lock in the +15 score.
ACTION: Place stone 2 in the cauldron. |
| Trial 5 Step 5 | GAME STATE UPDATE:
What stone do you use? 0
What potion do you use? 0
Outcome stone: blue small round with reward -1
New observation
You see the following stones:
0: blue small round with reward -1
1: None
2: None
You see the following potions:
0: None
1: orange
2: red
3: pink
4: None
5: red
6: red
7: None
8: red
9: red
10: orange
11: pink

Current trial score: 30

MODEL RESPONSE:
OBSERVATION: The Orange potion had no effect on the 'blue small round' stone. This confirms my hypothesis that Orange potions only work on 'pointy' stones.
REASONING: My only remaining stone is 'blue small round' with a reward of -1. I cannot improve its properties towards the maximum reward state because I lack the necessary Yellow and Turquoise potions. Placing it in the cauldron would lower my score. I have many Red potions, and my current hypothesis is that they change 'blue' stones to 'purple'. Since I cannot improve this stone, I will use this opportunity to test the effect of a Red potion on it to gather more data.
ACTION: Place stone 0 in potion 2. |
| Trial 5 Summary | * **Green Potion:** Changes a stone's color from 'purple' to 'blue'. The reward change depends on the stone's shape.
* If the stone is 'round', its reward increases by 2.
* If the stone is 'pointy', its reward increases by 14.
* **Red Potion:** Changes a stone's color from 'blue' to 'purple'. This transformation consistently subtracts 2 from the stone's reward.
* **Yellow Potion:** Changes a stone's shape from 'round' to 'pointy'. This transformation consistently adds 2 to the stone's reward.
* **Orange Potion:** Changes a stone's shape from 'pointy' to 'round'. This transformation subtracts 2 from the stone's reward.
This potion has no effect on 'round' stones.
* **Turquoise Potion:** Changes a 'purple small' stone into a 'purple large' stone. This transformation adds 2 to the stone's reward.
It appears to have no effect on stones that are already 'large' or are 'blue'.
* **Pink Potion:** Has no effect on a stone's properties or reward value.
### Highest Reward Combination
The single combination of stone properties with the highest reward is:
**blue large pointy**
This specific combination results in a reward value of **+15**. This state is best achieved by transforming a 'purple pointy' stone with a Green potion to gain the +14 reward bonus. |

Table 8: Outputs of Gemini 2.5 for Trial 5 of Alchemy in the condition with summarization and no prior information.

### A.4.5 Effects of few-shot prompting

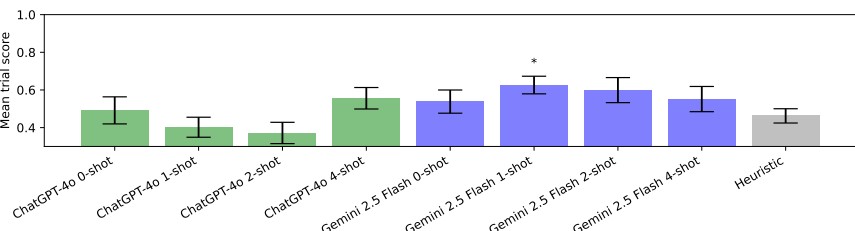

Figure 10: Mean normalized model scores for showing the effect of few-shot prompting on model performance. Scores represent the mean of 5 trials. Error bars represent standard error of the mean, across 10 replicates.

To test the effects of few-shot prompting on model performance in Alchemy, we collected trial histories from the best performing model, Gemini 2.5 Pro, to use as few-shot examples. Each example comprises all game state updates and Gemini 2.5 Pro responses (see Tables 7 and 8) for Trial 1 of an episode with a unique chemistry not present in the test set. We generated 4 such examples and appended 0, 1, 2, or 4 of them to the prompt for the test models. Examples were appended in a random order for each of 10 replicates. We used ChatGPT-4o (a model of similar size but lower performance) and Gemini 2.5 Flash (a model of smaller size) as test models. Both the examples and test model responses were generated in the condition with no summary and no prior information. To ensure that context limitations did not impact performance, we ran the test models for 5 trials rather than 10.

Our results show limited and inconsistent effects of few-shot prompting on model performance (Figure 10). Few shot prompting provided a slight improvement to scores for Gemini 2.5 Flash, but the size of the improvement did not increase with more than 1 shot. Interestingly, few shot prompting appeared to decrease performance for ChatGPT-4o in all but the 4-shot case. In all but the 1-shot case for Gemini 2.5 Flash, model performance was not significantly improved compared with the memoryless heuristic, suggesting limited utility of few-shot prompting for this task.

The limited effects of few-shot prompting may be due to the fact that the examples contain no information about the specific chemistry, so the models can only gain information on generally useful exploration strategies from them, the extraction of which is more abstract and difficult.

### A.4.6 Strategy adaptation timeseries results

### A.4.7 Effects of prior information ablation

To further investigate the effect of prior information in the prompt, we performed ablations in which we removed either reward information, potion pairing information, or causal information from the prompt with prior information (see Figure 9). We ran the models with the ablated prompt and no summaries enabled and measured the performance.

We found that the effects of prior information ablation differed substantially depending on the model. Gemini 2.5 is the most robust to the ablations, showing only a slight decrease in performance when no prior information is provided (Figure 12a). Interestingly, Claude 3.7 showed no decrease in performance with any of the single ablations, but still showed a large decrease when all prior information was removed (Figure 12b). This suggests that Claude 3.7 is more robust at meta-learning without summaries, but still struggles without at least a small amount of initial prior information to build off of. Performance of o4-mini was reduced slightly without causal and reward information, and moderately without potion pair information or when no information is provided (Figure 12c). Performance of ChatGPT-4o was reduced substantially and became statistically insignificant compared to the heuristic policy with removal of any piece of prior information (Figure 12d). This shows that this model is less robust at learning principles on its own, potentially due to its lack of thinking ability.

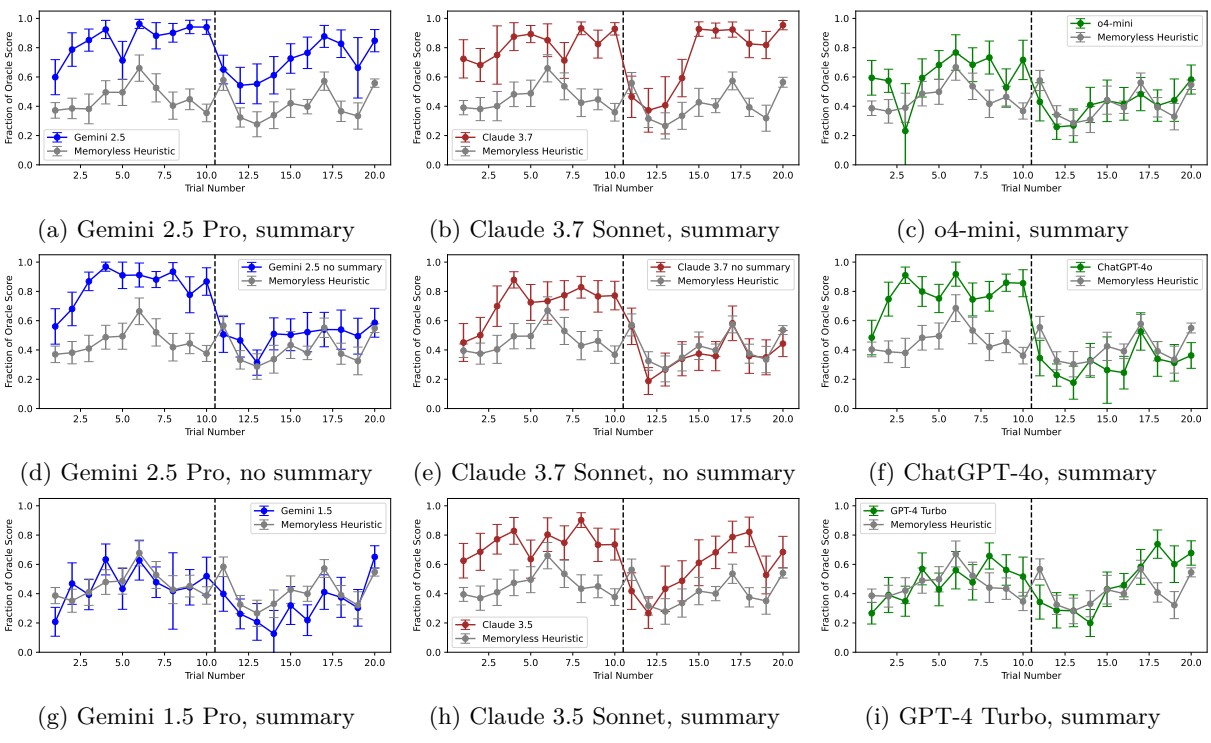

Figure 11: Normalized model score by trial when an uncued change in chemistry occurs halfway through the episode. (a-i) Trace of score across 20 trials. The vertical dotted line denotes the point at which the change in chemistry occurs, following trial 10. Error bars represent standard error of the mean, across 10 replicates.

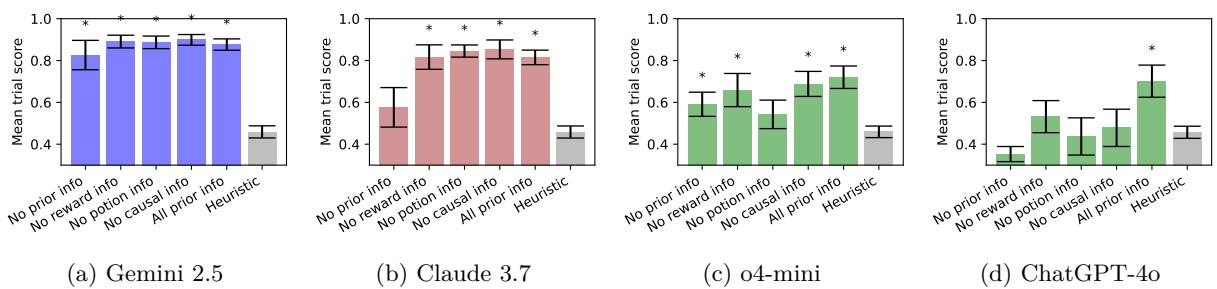

Figure 12: Mean trial scores for each model with no summarization when various pieces of prior information have been removed from the prompt. (a) Prompt ablations for Gemini 2.5. (b) Prompt ablations for Claude 3.7. (c) Prompt ablations for o4-mini. N=10 replicates of 10-trial episodes. Error bars represent standard error of the mean. Asterisk indicates the mean is significantly different from that of the memoryless heuristic ($p < 0.05$, paired-sample $t$-test).

### A.4.8 Alchemy item usage

To investigate the mechanism behind the score improvement in the models, we analyzed reduction in the number of potions used by the models in later trials (Figure 13). The most apparent finding is that Gemini 2.5 shows a significantly larger reduction in potions used than the other models when summaries are enabled. This demonstrates that the summary has a substantial effect on the model's strategy, despite the fact that the summary only improves the already-high performance of Gemini 2.5 a small amount. We also found that Gemini and Claude reduced their potion use substantially more than ChatGPT-4o overall, suggesting that these models owe part of their superior performance to an ability learn to make more efficient use of potions.

Finally, to analyze the propensity of the models for clear reasoning errors in different conditions, we computed the fraction of trials in which a model places at least one negative-valued stone into the cauldron (Figure 14). Such an action is obviously counterproductive, and is never performed by the memoryless heuristic. We found that all models place negative stones into the cauldron at least a small number of times, but in most cases in less than 10% of trials. Interestingly, however, in the prior information conditions, Claude places negative stones in the cauldron in between 20% and 25% of trials, despite being a high performing model in these conditions. It is not clear why this occurs, but it suggests there is room for significant improvement in Claude on this task.

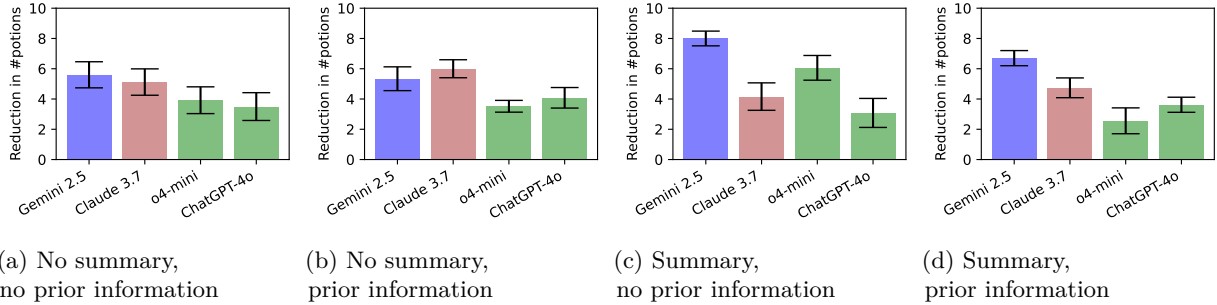

(a) No summary,
no prior information

(b) No summary,
prior information

(c) Summary,
no prior information

(d) Summary,
prior information

Figure 13: Reduction in the number of potions used over the course of the episode, computed as the potions used in the first episode minus the mean of the potions used in the last 5 episodes. (a-d) Same conditions as Figure 5. N=10 replicates of 10-trial episodes. Error bars represent standard error of the mean.

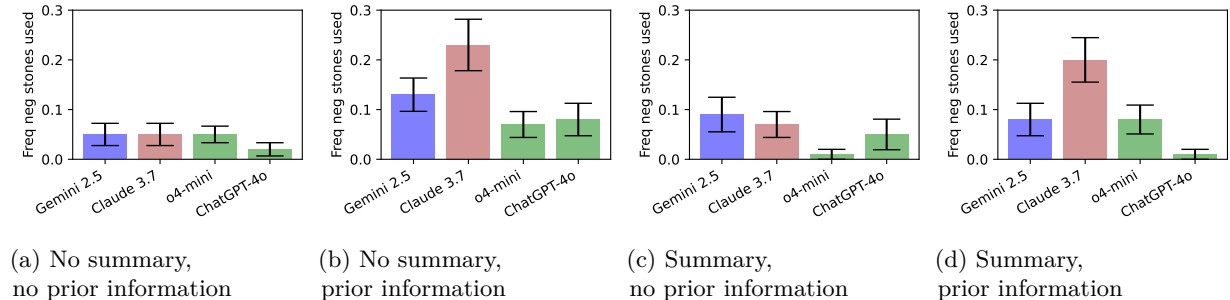

(a) No summary,
no prior information

(b) No summary,
prior information

(c) Summary,
no prior information

(d) Summary,
prior information

Figure 14: The fraction of trials in which at least one negative-valued stone was placed in the cauldron by the model. (a-d) Same conditions as Figure 5. N=10 replicates of 10-trial episodes. Error bars represent standard error of the mean.

