# OpenReview forum: "Exploring exploration with foundation agents in interactive environments"
_TMLR — Accepted by TMLR_

### Review · Reviewer_5UNj · 2025-09-11

**Summary Of Contributions:**

This paper systematically evaluates the ability of foundation models to conduct active exploration in interactive environments, focusing on three core capabilities: efficient information gathering, meta-learning, and strategy adaptation. The authors introduce two benchmark environments—a simple, text-based Feature World and a more complex, meta-learning-focused Alchemy—to test these abilities in a zero-shot setting across multiple state-of-the-art models, including various versions of Gemini, Claude, and ChatGPT. Key findings reveal that while most models perform near-optimally in simple information-gathering tasks, their ability to meta-learn and adapt strategies over time in complex, stateful settings like Alchemy is limited without explicit prompting. However, the study shows that prompting models to summarize information across trials unlocks emergent meta-learning and adaptation in some models, particularly Gemini 2.5 and Claude 3.7. A proof-of-concept 3D embodied version of Feature World also demonstrates that exploration capabilities can transfer to multimodal settings, though vision errors remain a performance bottleneck. The work provides valuable benchmarks and highlights both the current limitations and future potential of foundation models as autonomous exploratory agents. Strengths include its comprehensive evaluation and novel insights into eliciting meta-learning through summarization, while weaknesses involve the limited scope of 3D experiments and a focus only on in-context learning rather than fine-tuning.

**Audience:**

Yes

**Audience Explanation:**

LLM/vlm agents are currently a hot reasech topic in ml research, and understanding how agent interact with enviroment is interesting

**Claims And Evidence:**

Yes

**Claims Explanation:**

The claims in the submission are strongly supported by accurate, convincing, and clear evidence, as the authors employ rigorous experimental designs across multiple benchmarks (Feature World and Alchemy), test a range of state-of-the-art foundation models, and use appropriate baselines and statistical measures to validate their findings. Results are presented through well-labeled figures and tables that clearly illustrate performance trends, such as near-optimal efficiency in simple tasks and the emergent meta-learning capabilities enabled by summarization. The inclusion of ablation studies, error analysis, and transparent reporting of prompts and limitations further strengthens the validity and reproducibility of the conclusions

**Requested Changes:**

While the zero-shot focus is a strength, adding a small experiment (even in the appendix) comparing zero-shot performance to a few-shot  or a lightly fine-tuned model would greatly strengthen the paper. This could be done on the text-based Alchemy task.

---

### Review · Reviewer_sXfW · 2025-09-12

**Summary Of Contributions:**

## Summary

The paper evaluates foundation models as interactive agents in environments requiring multi-turn exploration.
- Authors introduce Feature World (text and 3D) to test information gathering about hidden reward functions. They then introduce to Alchemy, a multi-trial benchmark for meta-learning and strategy adaptation.


- Authors show that LLMs achieve near-optimal efficiency in simple Feature World tasks but struggle in Alchemy. They note that prompting models to summarize across trials unlocks emergent meta-learning and, in some cases, strategy adaptation.


- Authors benchmark multiple frontier models (Gemini 2.5, Claude 3.7, ChatGPT-4o, o4-mini), finding large gaps in robustness.




## Strengths
- Authors decompose key capabilities in exploration into three parts: information gathering, meta-learning, and strategy adaptation.
- Authors present interesting findings like summarization enables meta-learning; Gemini 2.5 shows strongest robustness.
- It’s novel to use Alchemy as a zero-shot exploration benchmark for LLMs capability, contrasting with previous RL results.


## Weakness
- The connection to related work on LLM’s capability on exploration, such as DISCOVERYWORLD: A Virtual Environment for Developing and Evaluating Automated Scientific Discovery Agents (arXiv:2406.06769) is not well discussed. It remains unclear what are the new insights this paper adds beyond those studies.
- Evaluation is limited to zero-shot prompting. The paper does not explore richer settings such as fine-tuning or integration with memory modules.

**Audience:**

Yes

**Audience Explanation:**

I think the audience working on the following domains would be interested: LLM for scientific exploration, LLM/VLM for meta-learing.

**Claims And Evidence:**

Yes

**Claims Explanation:**

I think the experiments are well structured, with baselines (e.g., optimal, random heuristic) and clear performance metrics. The writing is also clear.

**Requested Changes:**

See the first bullet in Weakness.

---

### Review · Reviewer_hSGY · 2025-10-19

**Summary Of Contributions:**

This paper proposes a few metrics to evaluate the performance of the exploration capabilities of LLMs (or LLM-based agents), including Efficient information gathering, Meta-Learning, and Strategy adaption. These three metrics are used to validate different perspective of LLMs' ability in exploring the environment and improving their own ability during exploring. After that, the author conducts a few experiments in some common datasets to calculate these metrics of different LLMs. The experimental results reveal some key differences in these metrics and indicate that LLMs can better handle the text-based task in the Efficient information gathering metric but usually fail to handle the complex multi-trail environment in the Meta-learning and Strategy adaption metrics. This study provides a fundamental exploration in the exploration ability of LLMs.

**Audience:**

Yes

**Audience Explanation:**

The finding of this paper is not so surprising; that is, it is not hard to see current LLMs will fall short in these complex tasks. A long time-scale continuous evaluation is urgently needed. However, such long-term evalution would be much beyond the scoop of this work. As a result, I would believe many people will be interested in the findings of this paper.

**Broader Impact Concerns:**

This work mainly introduces some interesting findings based on their experiments; so, there is no any concersn or ethical concerns in this work.

**Claims And Evidence:**

Yes

**Claims Explanation:**

I would agree that this paper's claim is supported by  accurate, convincing and clear evidence because the following reasons:
1. The author made claims based on their experiments instead of simply proposing some claims. That is, it has a "partially" rigrously defined metric, and they conducted experimetns to evalaute these metrics. The final results are based on the experimetal results.
2. While experiments are not sufficiently diverse and representative, it has been a good start for further exploring the related topics or abilities of LLMs.

**Requested Changes:**

1. I request the author to include a more rigours definition of each metric. For example, the efficient informatio gathering is calculated as the success rate within a fixed step budget, the author should give it a name or symbol, e.g. $\mathsf{M}\_{\text{info}}:= \frac{n}{m}$, where n represents ... and m represents ... . Put them in a separate section and list them with a clear definition.
2. On page 5, texts are overlapping each other.
3. What is "[reference-anonymized]"?

---

> ### Author Response · Authors · 2025-11-02
> **Added rigorous capability definitions**
>
> We thank the reviewer for their comments. We agree the paper would benefit from more formalized and rigorous definitions of the capabilities we measure. We address the three points raised by the reviewer:
>
> 1. We have revised the corresponding section in the introduction to include rigorous definitions as follows:
>
> *   **Efficient information gathering:** Selecting actions that maximally increase expected information gain. In Feature World, we operationally define this as the success rate ($R_{\text{success}}$) in finding a rewarding object within a fixed step budget $B$:
>
>     $$
>     R_{\text{success}} = \frac{1}{N} \sum_{i=1}^{N} \mathbb{1}(\text{steps}_i \leq B)
>     $$
>
>     where $N$ is the total number of episodes, $\text{steps}_i$ is the number of steps taken to find a reward in episode $i$, and $B$ is set to the maximum number of steps an optimal policy would need.
>
> *   **Meta-learning (learning to learn):** Improving expected performance on new tasks in a given family through experience of other tasks in that family (Thrun & Pratt, 1998). In Alchemy, we measure this as the mean within-episode score improvement ($I_{\text{score}}$) between the first trial and the average of the final trials:
>
>     $$
>     I_{\text{score}} = \frac{1}{N} \sum_{i=1}^{N} \left( \frac{1}{|T_{\text{late}}|} \sum_{t \in T_{\text{late}}} S_{i,t} - S_{i,1} \right)
>     $$
>
>     where $S_{i,t}$ is the normalized score in episode $i$ at trial $t$, and $T_{\text{late}}$ represents the set of final trials (trials 6-10 in our experiments).
>
> *   **Strategy adaptation:** Detecting when a strategy becomes invalid due to environmental changes and adapting by learning a new one. In Alchemy, we measure this as the mean performance recovered ($S_{\text{post}}$) following an uncued change to environment dynamics:
>
>     $$
>     S_{\text{post}} = \frac{1}{N} \sum_{i=1}^{N} \left( \frac{1}{|T_{\text{post}}|} \sum_{t \in T_{\text{post}}} S_{i,t} \right)
>     $$
>
>     where $T_{\text{post}}$ is the set of trials following the uncued change (trials 11-20 in our experiments).
>
> For each of the above capabilities, we say a model has that capability if the difference between the measured $R_{\text{success}}$, $I_{\text{score}}$, or $S_{\text{post}}$ for the model and the same measured quantity for a random or heuristic baseline is statistically significant ($p<0.05$).
>
> 2. We have fixed the overlapping text.
>
> 3. [reference-anonymized] is used because the actual reference could compromise the anonymity of our submission. It will be replaced with the actual reference in the camera-ready version.

---

### Decision · Action_Editor_2bTQ · 2025-11-24

**Recommendation:** Accept with minor revision

**Additional Comments:**

The paper was improved after the initial review, such as providing clarity on the metrics, among others. However, one reviewer suggested to give a proper discussion with related work like DiscovreryWorld. The newly added discussion still remained a bit abstract. Can you point to the specifi new insights that cannot come out of existing benchmarks, but are the findings in this work? Also, how are the findings here related to previous ones, eg, are they consistent and corroborate each other, or do we see conflicting signs?

**Audience:**

Yes

**Audience Explanation:**

Overall, reviewers find the work a useful contribution for researchers in large language models, automatic science discovery, AI in embodied environments, etc. The paper highlights limitations of current foundation models, and provides useful benchmarks and metrics to quantify them.

**Claims And Evidence:**

Yes

**Claims Explanation:**

This paper conducts systematic experiments to assess foundation models’ ability to gather information, meta-learning (learning to learn), and strategy adaptation (re-learning a world model when changes arise). It introduces two benchmarks: the simpler Feature World and the more complex Alchemy. Extensive experiments on some of the strongest models show their strong capability of information gathering, but major opportunities in meta-learning and strategy adaptation. The paper is well written. The details and transparency improves reproducibility and facilitate future work that builds on it. A technical limitation is that the work does not consider fune-tuning, which is out of the scope.